# Transcription Factor Action Orchestrates the Complex Expression Pattern of CRABS CLAW in Arabidopsis

**DOI:** 10.3390/genes12111663

**Published:** 2021-10-21

**Authors:** Thomas Gross, Annette Becker

**Affiliations:** Institute of Botany, Justus Liebig University, Heinrich-Buff-Ring 38, D-35392 Giessen, Germany; thomas_gross@t-online.de

**Keywords:** *Arabidopsis thaliana*, carpel development, complex expression pattern, transcriptional regulation, transcription factor

## Abstract

Angiosperm flowers are the most complex organs that plants generate, and in their center, the gynoecium forms, assuring sexual reproduction. Gynoecium development requires tight regulation of developmental regulators across time and tissues. How simple on and off regulation of gene expression is achieved in plants was described previously, but molecular mechanisms generating complex expression patterns remain unclear. We use the gynoecium developmental regulator *CRABS CLAW* (*CRC*) to study factors contributing to its sophisticated expression pattern. We combine in silico promoter analyses, global TF-DNA interaction screens, and mutant analyses. We find that miRNA action, DNA methylation, and chromatin remodeling do not contribute substantially to *CRC* regulation. However, 119 TFs, including SEP3, ETT, CAL, FUL, NGA2, and JAG bind to the *CRC* promoter in yeast. These TFs finetune transcript abundance as homodimers by transcriptional activation. Interestingly, temporal–spatial aspects of expression regulation may be under the control of redundantly acting genes and require higher order complex formation at TF binding sites. Our work shows that endogenous regulation of complex expression pattern requires orchestrated transcription factor action on several conserved promotor sites covering almost 4 kb in length. Our results highlight the utility of comprehensive regulators screens directly linking transcriptional regulators with their targets.

## 1. Introduction

Transcription is a universal process in which DNA is transcribed into mRNA that is exported from the nucleus and translated into protein sequence. Already, prokaryotes tightly regulate transcription for the proper timing of cellular development and metabolic processes. However, the prokaryotic way to control expression is different to eukaryotes, as co-functional genes are often grouped in co-regulated polycistronic operons (reviewed in [1]). In eukaryotes, genes involved in the same processes are distributed over the entire genome, such that every gene requires its individual regulatory sequence. Moreover, the promoter regions of eukaryotic genes are longer than those of prokaryotes, and they include more transcription factor binding sites, accession points for chromatin remodelers. Furthermore, distal regulatory elements such as enhancers or silencers can be many kilobases away from the transcription start site [2].

While the core promoter, which can contain a TATA box or an initiator element, enables the general expression of a gene by recruiting the basic transcriptional machinery [3,4], the fine tuning of expression is influenced by cis-regulating factors, such as enhancers, silencers and insulators (which block the action of distant enhancers and mark borders between hetero- and eu-chromatin), and by trans-regulating factors binding to these elements in the proximal and distal promoter [2,5]. Basal transcription factors act as pioneer factors, recruiting additional transcription factors, and opening up DNA-binding motifs for specific transcription factors [6]. The chromatin landscape surrounding the gene directly connects to the ability of transcription factors to bind DNA, such that histone tail modifications influence the accessibility of the chromatin [7,8]. Acetylation of histones, e.g., H4K16ac, leads to an opening of chromatin and a higher accessibility of DNA [7], while the trimethylation of H3K27 leads to condensation of chromatin resulting in reduced transcription, as in the *FLOWERING LOCUS C* (*FLC*) locus upon vernalization [9].

In addition, DNA methylation suppresses gene transcription. DNA methyltransferases add methyl groups to cytosine residues at three different motifs (CG, CHG, CHH) in plants. If present in promoter regions, DNA methylation usually inhibits enhancer binding and reduces expression [10]. The expression of *FLOWERING LOCUS T* is dependent on the distal enhancer Block C, while this block is usually demethylated, *FT* expression is inhibited when Block C is methylated [11].

Short interfering RNAs (siRNAs) and micro-RNAs (miRNAs) are responsible for RNA-dependent DNA methylation (RdDM) and post-transcriptional gene silencing (PTGS) [10]. While in RdDM siRNA activates de novo DNA methylation of complementary DNA regions, PTGS by miRNAs leads to the degradation of complementary mRNAs. Regulation by miRNAs occurs in multiple genes. For example, members of the HD-ZIP III family are regulated by miRNA 165/166, while miRNA172 binds to the *APETALA2* mRNA [12,13]. Regulation of gene expression is thus a combination of diverse regulatory modes, including miRNAs, DNA methylation, histone modifications, and transcription factor activity. However, the contribution of individual aspects of regulation are unknown for most genes. Even more so for genes regulating the development of complex organs that require precise spatial and temporal control of expression based mainly on internal signals. Additionally, while chromatin modifications, DNA methylation and miRNA binding sites can be measured with precision, transcription factor (TF) binding to specific DNA-binding motifs remains elusive. TFs bind to short (6–20 base pair) sequences, and these can occur frequently throughout the genome and be located at random positions. However, only small fractions of the sequences are bona fide targets of a particular transcription factor are bound [14], posing major challenges to distinguish the biologically relevant TFBS (TF binding sites) from those that simply match a factor’s binding specificity [14,15]. Experimental approaches to identify TFBS include ChIP-seq assays (chromatin immunoprecipitation-sequencing), which requires a TF-specific antibody to capture complexes including the TF bound to its target DNA [16]. However, this is conducted one TF at a time. To identify all TF’s regulating a single gene’s expression, only few experimental approaches are available, with Yeast One-Hybrid (Y1H) screens where a promoter sequence is used as bait against a TF library being the most extensively used [17]. Another option is to identify real TFBS in silico by searching for evolutionarily conserved sites, as these evolve more slowly than their flanking sequences [18]. Interestingly, even with abundant gene expression data, TF-binding data, and TF-expression data, we lack understanding how changes in TF activity causes changes in target-gene expression. Moreover, knowledge on the full set of TFs regulating the complex expression pattern of a developmental regulator in plants based on endogenous cues is unavailable.

The *Arabidopsis thaliana* protein CRABS CLAW is a member of the YABBY TF family, and *crc-8* mutants, such as the long-known *crc-1* mutant, have a shorter and wider gynoecium with the two carpels apically unfused, and they lack nectaries (Figure 1A–H) [19]. CRC specifies abaxial–adaxial polarity of the carpel, in concert with KANADI proteins and probably antagonistically to members of the HD-ZIP III protein family [20,21,22], and is involved in regulating floral meristem termination. CRC transcriptionally activates carpel target genes regulating nectary formation and gynoecium growth, and represses those involved in floral meristem termination [23]. *CRC’s* expression is strictly limited to the nectaries and the gynoecium (Figure 1E–H). In the gynoecium, it commences in stage six (stages according to [24]) in the gynoecial primordium and forms two distinct domains in the carpels after stages seven to eight: in epidermal expression around the circumference of the gynoecium, and an internal expression in four stripes that are close to the developing placenta [19,25]. The epidermal expression of *CRC* is consistent over the complete length of the carpels, but the internal expression forms a basal–apical gradient and ceases in later developmental stages. The epidermal expression is maintained until the middle of stage 12 in the valves, but it decreases earlier in the future replum. Expression in the nectaries starts at their inception and remains stable until after anthesis [19]. Previous analyzes of the *CRC* promoter by Lee et al. [25] identified five conserved regions (A–E) sufficient to drive *CRC* expression. Furthermore, Chip-seq data showed that the MADS box transcription factors SEP3, AG, PI, AP1, and AP3 bind to the *CRC* promoter suggesting their involvement in regulating *CRC* expression, such that *SEP3* together with *AG* as tetramers activate *CRC* expression, and *AP3* and *PI* repress *CRC* expression in the stamen and petal whorl [25,26,27,28,29,30]. Thus, previous work already provides a basic framework for activation and whorl-specific restriction of CRC expression.

However, we were interested in learning more about the regulation of the intricate *CRC* expression pattern in the gynoecium, as an example for a complex expression pattern of a developmental regulator independent of external cues. We combined experimental data with in silico analysis to identify putative regulators and their roles in *CRC* regulation. We find that TFs involved in several developmental pathways coordinate *CRC* expression via transcriptional activation, such as TFs directing flowering induction, floral organ identity and meristem regulation, and that most of them are only partially co-expressed with *CRC*. These regulators bind up to 3 kb upstream of the transcription start site of *CRC*, providing an example showing that complex expression patterns require long promoters

## 2. Materials and Methods

### 2.1. Plant Material and Plant Growth

All plants were grown on a soil–perlite mixture under standard long day conditions. For the crosses with the GUS reporter line, SALK lines of various transcription factors were used (Appendix A). For RNA in situ hybridizations, the SALK line (SALK_007052C, in Col-0) with a T-DNA insertion in the sixth exon of *CRC* (henceforth *crc-8*), the *half-filled*, *bee1*, *bee3* triple mutant (*hbb*) (a kind gift of Birgit Poppenberger and Martin Yanofsky), and the *cal* mutant (a kind gift of Daniel Schubert) were used. For a detailed description of *crc-8*, 100 randomly picked flowers at stage 14 [24] of *A. thaliana* Col-0 wild-type plants and *crc-8* plants, respectively, were manually dissected under a Leica M165C stereoscope (Leica Microsystems GmbH, Wetzlar, Germany) and analyzed (Figure 1A–H).

### 2.2. Y1H Assay

The *CRC* promoter (*proCRC*), as described by Bowman and Smyth [19] and Lee et al. [25], was amplified as a 3.8 kB fragment from genomic DNA of *A. thaliana* Ler-0 using the Phusion DNA polymerase (Thermo Scientific, Schwerte, Germany) in combination with the high-fidelity master mix (Thermo Scientific). The input amount of template DNA was 5 ng. Initial denaturing at 98 °C for 1 min was followed by 40 cycles of 10 s 98 °C, 20s 58 °C, 2 min 72 °C, and a final polishing step for 5 min at 72 °C. Additionally, the promoter was divided into seven fragments (*proCRC F1–F7*) and the five conserved regions (*proCRC A–E*) (Appendix A) that were identified by Lee et al. [25] were PCR amplified (for primers see Appendix A), digested with HindIII and KpnI, and cloned into the equally digested bait DNA vector pAbAi (Takara Clontech, Saint-Germain-en-Laye, France). The yeast strain *S. cerevisiae* Y1HG (Takara Clontech) was used for all Y1H analyses. The yeast transformation and autoactivation test was performed as described in [23]. The lowest Aureobasidin A (AbA) concentration that was sufficient to suppress yeast growth was used for the following screens (200 ng/mL AbA was used for all baits, except for the full-length bait in which 600 ng/mL AbA was used). AbA-sensitive strains were then transformed with the three prey libraries (for compositions of the three libraries see Appendix A and [31]). Prey plasmids from colonies > 2 mm diameter were isolated and sequenced. As positive controls, the yeast strains p53-AbAi + pGADT7-p53 and pKCS15-AbAi + pGADT7 CRC [23] were used with 100 ng/mL AbA and 150 ng/mL AbA, respectively)

### 2.3. Construction of proCRC:GUS Reportersystem and GUS Assays

As *proCRC* exhibits an internal BsaI recognition site, site-directed mutagenesis [32] of *proCRC* was performed to remove the BsaI recognition site (primers are listed in Appendix A) for the later integration of *proCRC* into the Greengate system [33]. After integration, the construct *proCRC:N-Dummy:GUS:C-Dummy:Ter_RBCS_;pMAS:Basta:Ter_MAS_* in the plant transformation vector pGGZ003 was assembled as described in Lampropoulos et al. [33]. The fully assembled vector was then transformed into *A. tumefaciens* GV3101 pSOUP+. These were transformed into *A. thaliana* Col-0 wild-type plants via floral dip as described in Davis et al. [34]. The resulting seeds were selected as described in [23]. Plants carrying *proCRC:GUS* were crossed with *A. thaliana* Col-0 loss-of-function mutants of putative *CRC* regulators. Young inflorescences of genotyped F2 plants were harvested in ice cold 90% acetone and incubated for 20 min at room temperature. The GUS staining and paraplast embedding was performed according to [35]. Then, 10 µm-thick sections from the embedded tissues were analyzed with a Leica microscope DCM5500.

### 2.4. Expression Analysis

RNA in situ hybridization to detect the *CRC* mRNA in carpel tissue was performed as described in [36] with modifications (see Appendix A). Probes were generated using T7 RNA Polymerase (for sequences see Appendix A).

*CRC* expression levels were analyzed via qRT-PCR in mutants of *CRC* regulators. Total RNA from buds of wild type, *arf8*, *agf2*, *athb16*, *bbx19*, *cal*, *cil1*, *ett*, *ful*, *haf*, *hat4*, *idd12*, *ino*, *jag*, *nf-y9*, *nga2*, *rve4*, *tmo5*, *ult1*, and *yab5* plants was isolated in quadruplicates using the NucleoSpin RNA Plant kit (Macherey-Nagel GmbH & Co., KG, Düren, Germany) and transcribed into cDNA using RevertAid H Minus Reverse Transcriptase (Thermo Fisher Scientific Inc., Schwerte, Germany) with random hexamer primer. A 1:50 cDNA dilution was added to the Luna master mix (NEB Inc., Frankfurt am Main, Germany) and the qRT-PCR was run on a Lightcycler 480 II (Roche Diagnostics Deutschland GmbH, Mannheim, Germany). *ACTIN2* was used as reference gene. Primer efficiencies were determined for *CRC* and *ACT2 2.1.* Primer sequences are listed in Appendix A. The raw data was analyzed using the Pfaffl method [37] and according to [38] (for detailed description see Appendix A).

### 2.5. In Silico Analysis of Genomic Loci, GO Enrichment and Co-Expression Analysis

The genomic loci of *APETALA2*, *FLOWERING LOCUS T*, *FLOWERING LOCUS C*, and *CRC* were screened for the presence of miRNA binding sites using psRNA Target (using standard settings) [39]. DNA methylation patterns were analyzed using data from the 1001 Arabidopsis Methylomes Project [40]. Histone modifications were identified with PlantPAN3.0 [41]. For functional categorization, the putative regulators were imported into Panther [42,43], and Gene Ontology terms were attributed from the GO biological process annotation data. Fisher’s Exact test was used and the Bonferroni correction for multiple testing with *p* < 0.05 was applied.

*CRC* co-expressed genes were identified via Pearson correlation using stage specific carpel (stage 5, 9, 11, and 12) RNA-seq data [44] and SAM, leaf, inflorescence, young flowers, and mature flowers RNA-seq data [45] based on their TPM (transcripts per million) values. Genes with positive correlation between 1–0.8 and with negative correlation between −0.8–−1 were used for further analyses [46]. Co-expressed genes present in the Y1H dataset were further analyzed in a heatmap generated with Heatmapper [47] using average linking and Pearson distances. The respective genes were scaled per row. BioGRID [48] was used to identify protein–protein interactions between the Y1H identified proteins for the assembly of co-regulatory networks.

## 3. Results

### 3.1. Regulation by DNA Methylation, Chromatin Modifications or miRNAs Plays Only a Minor Role in CRC Expression

*CRCs* expression is tightly regulated in a spatial and temporal manner, and is specific to carpel and nectary development (Figure 1I). We were first interested in understanding the contribution of the different means of transcriptional regulation of *CRC* expression. In an in silico approach, we searched specific databases for DNA methylation sites in rosette leaves, histone modifications, and miRNA binding sites in the *CRC* genomic locus (Figure 1I) and, in addition, analyzed the genomic loci of *APETALA 2* (*AP2*), *FLOWERING LOCUS T (FT)*, and *FLOWERING LOCUS C* (*FLC*). These genes are known to be regulatory by DNA methylation, chromatin modifications and miRNAs, respectively, and serve as controls.

The *AP2* genomic locus shows only a few DNA methylations between ~2 kB and ~1 kB upstream of the transcription start site (TSS), with five CHH methylations present and an additional CHH methylation at the end of the seventh exon. In contrast to this, multiple CG and CHH DNA methylations were identified ~1 kB upstream of the TSS of *FLC*. In addition, two CHG methylations were present ~1 kB and ~2 kB upstream of the TSS. The DNA methylation pattern in *FT* is more complex, as methylation marks concentrate on three regions. Approximately 0.5 kB upstream of the TSS are two CG methylation sites and one CHH, 5.5 kB upstream with only few DNA methylations (CG and CHH) in the *FT* promoter and in a highly methylated stretch of ~1 kB between 2.7–3.7 kB upstream of the TSS, including many CG and CHH methylations but also few CHG methylations. The *CRC* genomic locus shows two sites of CHH DNA methylations (~0.3 kB and ~3 kB upstream of the TSS), suggesting little influence of DNA methylation on its gene expression. However, DNA methylation acts dynamically, and our data relate only to rosette leaves.

Activating and repressive histone marks were found in the genomic loci of all four genes based on ChIP-Seq data from vegetative plant tissues. Both the genomic loci of *AP2* and *FLC* showed the highest number of histone marks and included activating and repressing marks, covering most of promoter regions and the coding sequences. In contrast, *CRC* and *FT* genomic loci show only two repressive marks, with H3K27me3 covering most of the genomic locus, including the promoter and H2AK121ub covering the transcribed region, suggestive of some degree of regulation by chromatin methylation and ubiquitination. MiRNA binding sites were identified only in the last exon of *AP2*.

In summary, our in silico analysis corroborates that *AP2* is regulated by all analyzed means of expression regulation. *FLC* shows regulation by activating and repressive histone marks as well as CHH, CG, and CHG DNA methylation. *FT* has only few types of repressive histone marks present and is regulated by extensive CHH, CG, and CHG methylation. In contrast, *CRC* is regulated independently of miRNAs and DNA methylation. Similar to FT, it shows only two types of repressive histone marks in vegetative stages of development.

### 3.2. Diverse Transcription Factors Bind to the CRC Promoter

Because we have shown that the genomic locus of CRC is not affected DNA methylation, only mildly by histone modifications, and that miRNA cleavage of its transcripts also plays no role in gene-expression regulation (Figure 1), we hypothesized that *CRC* is regulated mainly by transcription factors. To identify the direct upstream regulators of *CRC*, a Yeast-1-hybrid (Y1H) screen of the *CRC* promoter was performed in which the full-length promoter (3.8 kb) and 12 smaller promoter fragments (Appendix A) were transferred as baits into yeast. Three bait strains (*proCRC A*, *proCRC F2*, and *proCRC F5*) were discarded because they showed autoactivation with resistance to 1000 ng/mL AbA. The remaining ten strains were transformed with the three different libraries of prey TFs (Appendix A and [31]) and grown on selective SD-Leu or SD-Trp medium. A total of 140 proteins binding to the *CRC* promoter fragments in yeast were identified (Appendix A). We used the PlnTFDB [49] and TAIR databases to assign these TFs to a protein family. To identify those that bind sequence-specifically, we searched PlantPAN 3.0 [41] for the presence of their experimentally verified DNA-binding motifs in the *CRC* promoter (Appendix A). Notably, Y1H screens can only identify proteins that bind as monomers or homodimers/homomultimers to the bait promoter.

A total of 34% (48 proteins) of the 140 proteins are present in the PP database and have a DNA binding-motif match within the *CRC* promoter sequence (Figure 2A). The remaining 92 prey proteins could be separated into two categories: (1) 71 proteins (51%) were not included in PP, because their DNA-binding motif is unknown, (2) 21 proteins (15%) were included in PP but their binding motifs do not match to the CRC promoter. The 21 proteins from the second category may have additional binding motifs not yet identified, or they have no binding site in *proCRC* and can be seen as false positives and were thus excluded from further analyses. All genes encoding proteins identified in this Y1H screen are expressed during gynoecium development [45,50] and resemble multiple protein families (Appendix A). The proteins binding to the *CRC* promoter in yeast include well-known carpel developmental regulators such as HALF FILLED (HAF), FRUITFUL (FUL), ETTIN (ETT) and ARF8. However, also genes so far not known to act in the gynoecium, such as REVEILLE 4 (RVE4), required for circadian rhythm maintenance and response to heat shock [51,52], or WRKY41, involved in regulation of anthocyanin biosynthesis [53] (Figure 2A). These results suggest that *CRC* expression regulation relies on several, so far seemingly unrelated, developmental pathways.

### 3.3. Relevant Promoter Fragments Are Enriched in TFBS and CRC Regulators Are Functionally Related

We were then interested to see if the binding sites of the direct regulators are located in the regions conserved between Brassicaceae [25]. We plotted the number of TF binding sites per 100 bp in one region identified via in silico prediction (light grey) and identified in the Y1H screen (dark grey) (Figure 2B). On average, 6.37 and 8.52 binding sites per 100 bp were identified, fragments B (9.13 and 19.13, respectively) and C (7.28 and 18.35, respectively) show the most binding sites, with an additional maxima in A for the in silico identified binding sites (8.49 and 5.66, respectively). Overall, the fragments including the fewest putative binding sites are all between the conserved blocks identified by [25], corroborating their results from promoter shading and promoter-GUS assay analysis.

We were then interested to learn about the function of the *CRC* regulators. Thus, the 119 putative *CRC* regulators (those identified with Y1H and known DNA-binding motifs in *proCRC* plus those of category (1), were divided into functional groups based on gene ontology terms (Figure 2C). Ten GO terms were overrepresented among the candidate genes (abaxial cell fate specification, carpel development, meristem maintenance, regulation of flower development, positive regulation of transcription by RNA polymerase II, vegetative to reproductive phase transition of meristem, regionalization, response to auxin, negative regulation of transcription (DNA-templated), and response to light stimulus), while three GO terms were underrepresented (cellular metabolic process, unclassified, and protein metabolic process). Most enriched terms are closely related to known functions of *CRC*, in particular, the terms abaxial cell fate specification (GO:0010158) and carpel development (GO:0048440) are highly enriched, with a 6.47 log 2-fold enrichment and 4.28 log 2-fold enrichment, respectively. Only the weakly enriched (2.26 log 2-fold) response to light stimulus (GO:0009416) is not directly related to *CRC* functions but might be connected to light-induced flowering, through genes such as flowering time regulators *LIGHT-REGULATED WD2* (*LWD2*) and *VASCULAR PLANT ONE ZINC FINGER PROTEIN 2* (*VOZ2*).

### 3.4. CRC Expression Is Activated by Diverse Developmental Regulators

We were then interested in how the *CRC* regulators influence the pattern and strength of CRC quantitatively *in planta*. *CRC* expression in *hbb* and *cauliflower* (*cal*) was visualized using mRNA in situ hybridization and showed no differences to the wild-type expression (Appendix A). Interestingly, no *CRC* expression was found in the newly characterized *crc-8* (Figure 1, Appendix A), suggesting autoactivation or maintenance of *CRC* transcription. Further, we chose four mutants (*agf2*; *Arabidopsis thaliana homeobox 16*, *athb16*; *bbx19*; *indeterminate domain 12*, *idd12*; *inner no outer*, *ino*) based on their defects in flower development or phytohormone signaling (Appendix A) for GUS staining assays. The localization of *CRC* expression in all four mutants was similar to the wild type, suggesting that the loss of function of those single regulators has no effect on the spatiotemporal expression of *CRC* (Appendix A). All observed mutants showed the phenotypes previously described [39,52,54,55,56,57,58,59,60,61].

We also quantified changes in *CRC* expression in 20 homozygous regulator mutants via qRT-PCR (Figure 3A). *CRC* transcript levels in buds are significantly reduced in 14 mutants: *agf2*, *bbx19*, *cal*, *ettin* (*ett*), *fruitful* (*ful*), *half-filled bee1 bee 3* triple mutant (*hbb*), *hat4*, *idd12*, *jagged* (*jag*), *nf-ya9*, *ngatha2* (*nga2*), *reveille4* (*rve4*), *ultrapetala1* (*ult1*), and *yabby5* (*yab5*) (Figure 3A). Of those, *CRC* expression was decreased by only ~25% in *hat4* (0.78 ± 0.11), *idd12* (0.78 ± 0.16), *rve4* (0.74 ± 0.15), and *ult1* (0.74 ± 0.15) when compared to wild-type expression. The other regulator mutants showed *CRC* expression reduction between 25–50%, for example in *ett* buds, *CRC* expression was only half as strong as in wild-type buds (0.46 ± 0.12). It is noteworthy, that some of these mutants show abnormal gynoecia, such as *ett* which shows defects along the apical–basal axis, and *hbb* shows defects in transmitting tract development late in gynoecium development [54,56]. However, for *nga2* and *cal* single mutants, no phenotypes in the gynoecium were reported previously [58,62]. Our findings together with published data indicate that at least some of these transcription factors activate expression of *CRC* in the gynoecium and nectaries, even though one cannot rule out that the downregulation may, to some extent, be due to gynoecium tissues being absent due to mutations present in these plants.

*Auxin response factor 8* (*arf8*), *athb16*, *cib1-like protein 1* (*cil1*), *ino*, and *target of monopteros 5* (*tmo5*) mutants showed *CRC* transcript abundance similar to the wild type (Appendix A). This suggests that 6 proteins out of 20 bind to the *CRC* promoter in yeast and have predicted binding sites in this promoter, but do not contribute to *CRC* expression regulation, while 14 proteins activate *CRC* expression. Among the activators are at least four (AGF2, BBX19, CAL, HAF) that, as single genes, have no influence on the pattern of expression but do on transcript abundance.

### 3.5. Regulators of CRC Are Partially Co-Expressed during Flower Development

As several methods for TF target prediction use co-expression of the TF and its target genes (e.g., [64]), we were interested to learn how useful this method is for identifying regulators of complex expression patterns. We thus carried out digital gene-expression analysis of the candidate regulators to discriminate genes with expression patterns similar to *CRC* from those with complementary expression patterns. A total of 7577 genes were co-expressed with *CRC* based on Pearson’s correlation (correlation coefficient between 0.8–1 and −0.8–−1), and among those, 5167 were positively and 2410 negatively correlated with *CRC* expression. Further, 555 of the co-expressed genes were TFs, including 381 positively and 174 negatively correlated genes. A total of 32 co-expressed genes encoded transcription factors binding to the *CRC* promoter shown in the Y1H experiments (Figure 2, Appendix A). To these 32 co-expressed genes we added the candidate regulators chosen for mutant analysis, if they were not already present (Figure 3B). These co-expressed genes assemble into seven groups: groups one to four include genes that are mainly expressed in late carpel stages and include for example *KNAT1*. Groups five to seven genes are highly expressed in early carpel stages. Genes expressed mainly in early carpel developmental are members of groups five to seven, including CRC and most other genes encoding for proteins shown to interact with proCRC in the Y1H screen, including well-known genes such as *ETT*, *BHL9* (*RPL*) and *ULT1*. Group four members are mainly expressed in the latest stage of carpel development. Group two and group three members are most strongly expressed in the carpel at stage 11 of flower development and include genes such as *JAG*, *ARF8*, *FUL*, *NGA2*, and *SEP3,* and also four genes with almost exclusive expression in stage 11 including *HAF*, *INO*, and *CAL*. Using additional RNAseq data from leaves, SAM, inflorescences, young flowers, and mature flowers further subdivided the four categories into seven (Appendix A)

In summary, we find that the almost half of the proteins interacting with the *CRC* promoter show an expression pattern very similar to that of *CRC,* indicating a role in activation or maintenance of *CRC’s* expression. Other groups (one to four) show partially or fully complementary expression patterns, suggestive of a function in negative regulation of *CRC*.

## 4. Discussion

### 4.1. A Complex Interplay of Transcription Factors Regulates CRC Expression

*CRC* is expressed in a complex spatial and temporal manner, and as *CRC* expression is not regulated by DNA methylation, miRNAs or repressive histone marks (Figure 1I), its spatial and temporal expression regulation relies mainly on TF networks (Figure 4A).

*CRC* is integrated in different regulatory networks necessary for flower development, such as the termination of the floral meristem or adaxial/abaxial polarity of carpel development [19,59,66] (Figure 4B). The members of gene regulatory networks are often co-expressed, and co-expression analysis may be used to carefully predict the function of a gene, as co-expressed genes are not necessarily cofunctional [67]. Here, we combine data from the co-expression analysis and Y1H screen (Figure 2 and Figure 3) with data derived from the literature to maximize the likelihood of finding “real” regulators of *CRC*.

This work focuses more on identifying novel transcription factors that may regulate the intricate expression pattern of *CRC* in the gynoecium. Among the regulators co-expressed with *CRC* in the gynoecium and a strong expression in the SAM is AGF2, which activates *CRC* when FUL expression is low (Figure 3A,B). Additionally, GIANT KILLER, an AT-hook-type DNA binding protein, which is known to counteract *ULT1* activity, [65] shows this peculiar expression pattern. GIK expression is directly activated by AG and it is known to regulate genes involved in carpel development such as *ETT,* by adding the repressing histone marks H3K9me2 to the *ETT* promoter [65]. GIK binds *proCRC* fragments C, D, and E and may repress *CRC* expression in late stages of gynoecium development in a way similar to *ETT.*

ULT1, a SAND and trithorax domain containing transcriptional regulator [68], activates *CRC* expression in flowers (Figure 2 and Figure 3A). It mediates the removal of repressive histone H3 lysine methylation marks (H3K27me3) or hinders their new positioning to activate the expression of its target genes, such as *AG* [60]. As ULT1 binds to the *CRC* promoter regions B to C, it may mediate the removal of repressive histone marks from the *CRC* genomic locus early in gynoecium development, in a way similar to that which was shown for *AG* [60]. *ULT1* also indirectly activates *CRC* by activation of *AG*, which, in turn activates *CRC* expression [19,25,27]. Interestingly, *CRC* and *ULT1* act redundantly to terminate the floral meristem [59] suggesting that *ULT1* and *CRC* act on the same targets while *CRC* itself is a target of *ULT*.

ETT binds to the promoter fragments A to E in yeast (Figure 2A), and its loss of function results in the strongest decrease of *CRC* expression when compared to all other genes tested (Figure 3A), suggesting that ETT is an important activator of *CRC* transcription. In leaves, ETT activates the expression of the YABBY genes *FIL* and *YAB3* which act in combination with KAN genes in specifying abaxial polarity [69]. Because in carpels it is *CRC,* which is involved in abaxial polarity specification, *ETT* may target YABBY genes in a more general way. Interestingly, *ETT* is only weakly co-expressed with *CRC* and provides an example of an important transcription factor not strongly co-expressed with its target.

Interestingly, several genes activating *CRC* expression are only weakly co-expressed with *CRC*. Among those are *FUL*, *HAF*, *NGA2*, and *JAG*, which are all most strongly expressed in later stages (Figure 3B) of carpel development but strongly activate *CRC* expression (Figure 3A) and bind to *proCRC* (Figure 2A). The bHLH protein HALF FILLED is necessary for transmitting tract development to enable the pollen tubes’ growth for ovule fertilization [56,70]. FUL acts antagonistically to RPL and together they determine valve identity and are necessary for the elongation of the developing fruit [55,71]. A ChIP-SEQ analysis of FUL targets did not identify *CRC* [72], but only gynoecia and fruits after stage 12 were part of this analysis, suggesting that FUL may be absent from the *CRC* promoter in late stages of carpel development. NGA2 binds to regions E and A of *proCRC*, and it participates in the formation of style and stigma and is involved in longitudinal growth of the gynoecium [58,73]. As *crc* gynoecia are typically shorter than those of the wild type, NGA2 might act via activating *CRC* to control this longitudinal growth.

Additionally, JAGGED (JAG) activates *CRC* expression while it is only weakly expressed during early gynoecium development, and it interacts genetically with several co-expressed proteins. This group of genes may act in a concentration-dependent regulation such that FUL, HAF, NGA2, and JAG activate *CRC* at low protein concentrations, and repress *CRC* at high protein concentration.

As expected, MADS-box proteins are involved in *CRC* regulation: SEP3 and SEP4 physically interact with AG, AP1, and PI, which are known to regulate *CRC* [19,27]. Thus, their binding sites can serve as hubs for MADS regulation. Additionally, CAL binds to *proCRC* in its C region, but is not known to have roles late in flower development. It acts redundantly with *AP1* to orchestrate the transition from inflorescence meristem to floral meristem and is expressed mainly in the floral meristem, sepals and petals [4]. However, *CAL* is also necessary to activate other flower developmental genes and may be involved in the initiation of *CRC* expression at low dosage (Figure 3A). In addition, *CAL* may have a repressive function on *CRC* late in gynoecium development when it shows a peak of expression at stage 11 (Figure 3B) suggesting a dosage-dependent action on *proCRC*. As the Y1H approach allows only identification of proteins binding to DNA as homodimers or homomultimers, it is not surprising that AG, AP3, and PI were not identified as interactors, because they require other proteins, such as SEP3 for AG and AP3 for PI (and vice versa) for their actions as transcription factors [30,74].

*CRC* expression is regulated by members of different developmental networks connected by protein interactions, which are all involved in reproductive development (Figure 4B): the genetic network regulating floral induction by light with CO as a central regulator acts on *proCRC* directly via RVE4, BBX19, and NF-YA9. The floral meristem regulation network acts on *proCRC* via JAG, IDD12 and GIK, and the floral organ identity network via CAL, FUL, and AGF2. CRC only directs auxin synthesis and readout by repressing *TRN2*, a modulator of auxin homeostasis, and by regulating *YUC4*, an auxin biosynthesis gene [5], but our data show that it is also regulated by the auxin response factor ETT, HAT4, and NGA2, the latter being involved in auxin signaling (Figure 4B). Additionally, adaxial/abaxial polarity regulators such as ETT, ULT1, and YAB5 activate *CRC* expression, as well as HAF, which is a gynoecium morphogenesis regulator. Proteins described as members of the gynoecium morphogenesis network [63] are also participating in other networks (Figure 4B), such as SEP3, FUL, ETT, NGA2, KNAT1/BP, or RPL regulate *CRC* expression, suggesting that the *CRC* promoter receives signals from several interconnected developmental GRNs allowing precise timing, spatial distribution, and control of transcript abundance for proper *CRC* expression.

### 4.2. Regulation of Complex Expression Patterns

The regulation of a single gene’s expression at the level of timing, distribution, and abundance of transcripts in a comprehensive way is addressed in surprisingly few studies. For genes responding to external cues such as heat stress, interaction between auto- and cross regulation of TFs, epigenetic and post-transcriptional regulation is combined to acquire thermotolerance and long-term adaptation to heat stress (reviewed in [75]). However, these genes are expressed rather uniformly in the plant or at the site of induction. In contrast, developmental regulators respond to internal cues, such as regulation by other transcription factors, and for some of them, one being *CRC*, these internal cues of assumedly several pathways are summed up and produce intricate mRNA patterns in space and time. Many examples of these developmental gene regulatory networks (GRNs) are known, e.g., for flower development [76], or leaf development [77] or the precise spatial patterning of lignification allowing for fruit dehiscence [78]. However, most of these studies used co-expression analyses or reconstructed GRNs by assembly of single/few genes’ genetic interactions. In contrast, our study provides evidence independent of genetic interaction or co-expression studies on the complex regulation of complex expression patterns of a key developmental regulator. Our work indicates that the combinatorial action of TFs is important for patterning of expression, but less so for the regulation of transcript abundance. This leads to the question of if TF’s transcript abundance is of lesser importance than the spatial distribution of transcripts. Many genes are thought to be differentially expressed with a log 2-fold change difference [79] between treatments or developmental stages. However, one may ask how relevant this threshold is for developmental regulators, which may act at low abundance. However, other TFs such as WUS or PLE are known for their dosage dependence, as high transcript/protein abundances result in binding of low-affinity binding sites and low protein abundances result in high-affinity binding-site usage (reviewed in [80]).

GRN members generally seem to be co-expressed, and this knowledge is used to link unknown genetic connections to GRNs in silico. However, our work shows that this approach may be used only with utmost care, as most RNA-seq data (e.g., [45,81]) lack the precision for taking tissue- or cell-type-specific expression into account, or are not developmental stage-specific. Further, many developmental regulators act in a tissue context-dependent way, together with different protein interaction partners, and on several target genes. For example, the *CRC* activator ETT interacts with TEC1 to repress the growth of side shoot structures in an auxin-dependent manner [82] and interacts with ABERRANT TESTA SHAPE (ATS or KAN4) to define boundaries between integument primordia of ovules [83]. Within the tissues selected for the co-expression analysis (Figure 3B), *ETT* and *CRC* are co-expressed, but if the inflorescence axis, hypocotyl, and ovules would have been added, together with the ovule-free carpel datasets from [63], co-expression between the two genes would be difficult to find, as *ETT* shows expression in those tissues [45] but *CRC* does not (Appendix A). The same is true for *ATHB16*, a gene that is co-expressed with *CRC* in our datasets but does not regulate *CRC* (Appendix A) and would most likely be a false positive member of the *CRC* containing GRN. Conversely, *CAL* is hardly expressed in the gynoecium but activates *CRC* expression (Figure 3A). This shows that co-expression and co-regulation are not necessarily linked and that the assembly of GRNs based on co-expression networks requires extensive experimental verification.

## 5. Conclusions

While we tried to obtain a comprehensive picture of experimentally verified, direct regulators of *CRC* expression, the Y1H approach identifies only transcription factors acting as monomers or homodimers. To our knowledge, there is no published information on the ratio of TFs forming homo- vs. hetero-dimers in plants. However, a compilation by [63] shows that among the known interactions of carpel development regulators, only 25 can homodimerize but 56 cannot, and that dimeric interactions change over time. If TFs can bind to DNA only in a specific combination with another TF, Y1H will not identify this interaction. Once a homodimer is identified, higher levels of complexity can be analyzed by a combination of protein interaction data resulting from other methods, such as Yeast Two-Hybrid (Y2H) or CrY2H-seq [84] (Figure 4).

Our data further show that transcription-activating TFs, such as ETT, bind up to 3 kb upstream of the transcription start site (Figure 2, Appendix A, [25]). These experimental findings do not corroborate in silico analyses [85], showing that the majority (~80%) of TFBS of the majority of promoters are between −1000 and +200 of the transcription start site. However, genes with complex expression patterns may be also the exception to the general positional preferences with more extended promoters. In addition, many methods used to identify transcription factor DNA-binding motifs take only monomers or homodimers into account and thus show often palindromic binding motifs. However, the formation of heterodimers can influence directly the DNA-binding motifs of the two dimerizing proteins [86,87], thus increasing the difficulty of in silico binding site predictions.

In summary we can state that the comprehensive analysis of factors regulating complex transcription factors is, at the current state of wet lab and in silico methods, challenging. Directed manipulation of developmental regulators’ expression patterns for yield improvement is thus difficult to achieve and requires extensive research.

## Figures and Tables

**Figure 1 genes-12-01663-f001:**
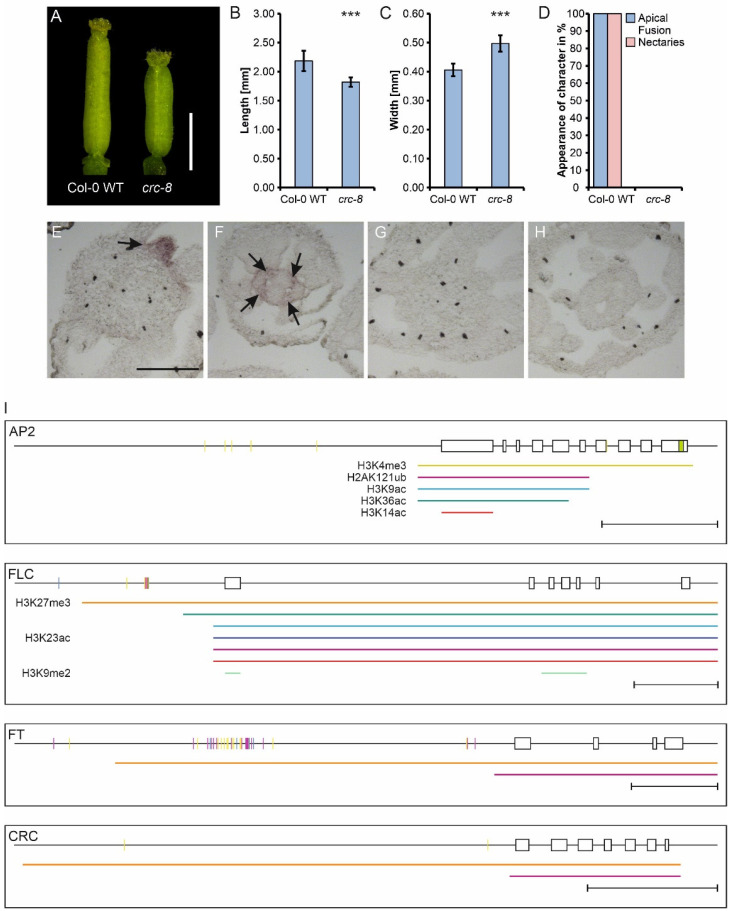
*Arabidopsis thaliana crc-8* phenotype and summary of gene-expression regulation of *AP2*, *FLC*, *FT*, and *CRC*. (**A**) Representative gynoecia of Col-0 wild type and *crc-8* plants. Scale bar represents 1 mm. Statistical analysis of gynoecium length (**B**), width (**C**), and a summary of absence or presence of other described *crc-1* phenotypes in *crc-8* (**D**). Both length and width comparisons (**B**,**C**) are the means with standard deviation. Percent values are shown in (**D**). Student’s *t*-test was applied to compare the wild-type gynoecia with *crc-8* and significant differences were marked with up to three asterisks (*p* < 0.001), *n* = 100. (**E**–**H**) Spatial analysis of *CRC* expression with RNA in situ hybridization. In situ hybridization using a *CRC* antisense probe of *A. thaliana* Col-0 wild type (**E**,**F**) and *crc-8* (**G**,**H**), showing gynoecia (**E**,**G**) and nectaries (**F**,**H**). Nectaries and internal *CRC* expression marked with arrows. (**I**) Summary of gene-expression regulation of *AP2*, *FLC*, *FT*, and *CRC*. Shown are the promoter regions and the exon or intron structure of the respective gene with exons shown as boxes. DNA methylation is shown in short purple (CG), blue (CHG), and yellow (CHH) vertical lines. Colored horizontal lines under the genomic locus indicate regions of histone modifications identified with PlantPAN in leaf tissue: the activating marks H3K4me3 (yellow), H3K9ac (light blue), H3K14ac (red), H3K23ac (dark blue), H3K36ac (dark green), and the repressing marks H2AK121ub (magenta), H3K9me2 (light green), and H3K27me3 (orange). Sorting into activating or repressing marks was performed according to [1,2]. miRNA binding is indicated by a green box in the respective exon. Scale bars represent 1 kB. The ChIP-Seq data used for histone mark identification resulted from only vegetative plant material (seedlings, leaves, roots, and shoot apical meristems or young inflorescence meristems), thus resembling only the state of histone modifications in these tissues.

**Figure 2 genes-12-01663-f002:**
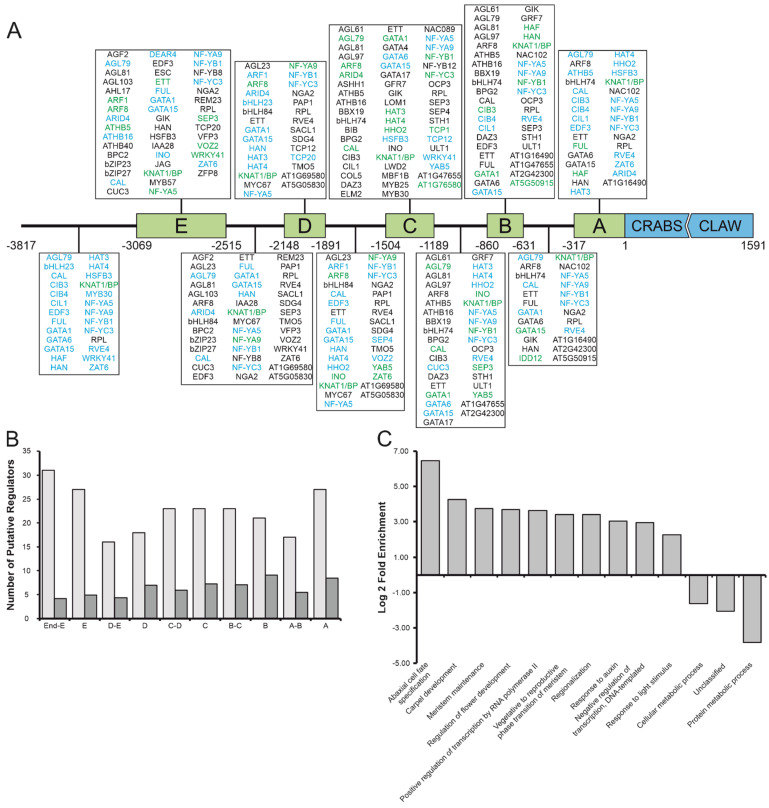
Analysis of transcription factors binding to the *CRC* promoter identified by Yeast One-Hybrid analysis. (**A**) Spatial distribution of transcription factor binding sites summarizing the Y1H screen of *proCRC* with transcription factor bait libraries using the fragments indicated in Appendix A as prey. Shown are only transcription factors with a known motif in PlantPan. Binding site proteins in black were identified by Y1H, those in blue by PlantPAN in silico prediction, and those in green indicate positional overlap of Y1H and PlantPAN data. (**B**) Quantitative analysis of putative CRC regulators’ distributions across the different fragments of *proCRC*. The number of regulators identified by in silico prediction with PlantPAN per 100 bp is shown in light grey, and the number of transcription factor binding sites identified by the Y1H screen per 100 bp are shown in dark grey. (**C**) GO enrichment analysis, categorizing the putative CRC regulators in different functional groups. Shown is the log 2-fold enrichment of significantly overrepresented GO terms.

**Figure 3 genes-12-01663-f003:**
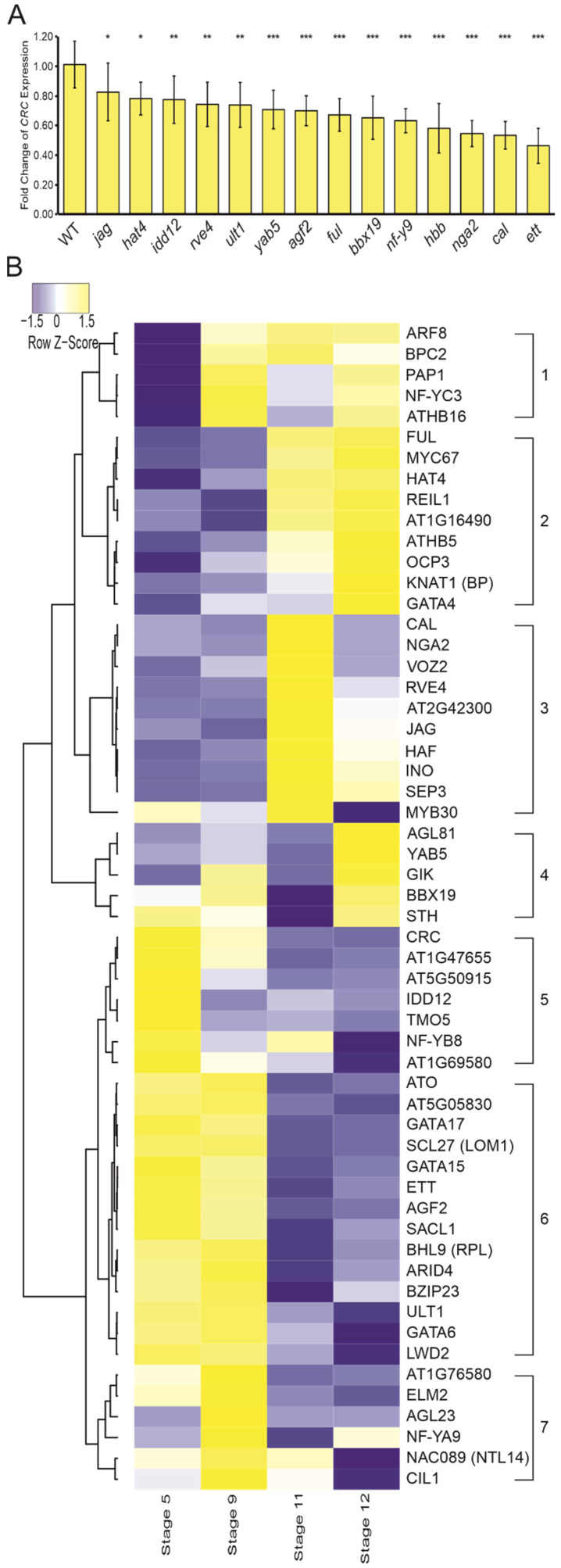
Quantitative effect of regulator mutants on *CRC* expression and digital gene expression analysis of putative *CRC* regulators. (**A**) *CRC* expression in candidate mutant lines in relation to *CRC* expression of Col-0 wild-type buds given as mean values of the fold change of CRC expression, error bars indicate standard deviation. (**B**) Heatmap of *CRC* and co-expressed putative regulators during four carpel developmental stages [63], showing correlation of expression by hierarchical clustering. Color intensity represents z-score. * *p* < 0.05, ** *p* < 0.01, *** *p* < 0.001.

**Figure 4 genes-12-01663-f004:**
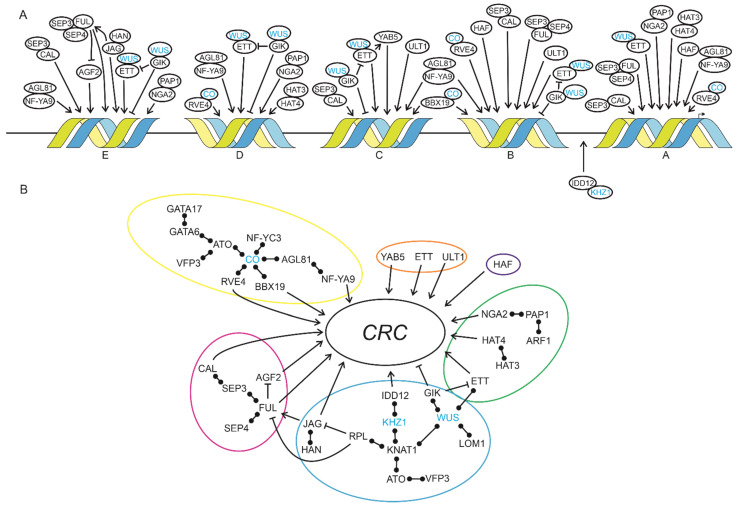
Networks regulating CRC expression. (**A**) Localization and binding of CRC regulators based on Y1H and qRT-PCR results and [65]. Regulators were assigned to the conserved regions of proCRC (A–E) accordingly to the Y1H screen and in silico prediction analyses. Proteins without an arrow were not tested via qRT-PCR but interact with proteins identified in this study based on BioGRID searches [3]. Protein–protein interactions are symbolized by overlapping circles and proteins in blue were not present in the Y1H results, but link identified regulators to each other. Protein interactions are shown by overlapping circles, transcriptional activation and repression are symbolized by pointed or blunt-end arrows, respectively. (**B**) Contribution of flower-related processes to CRC expression. Colored circles indicate cofunctional networks with flowering induction in yellow, ab-/adaxial regulation in orange, carpel structures in dark blue, auxin response in green, meristem regulation in light blue, and flower development in purple. Protein interactions are shown by bars with two circles, transcriptional activation and repression are symbolized by pointed or blunt-end arrows, respectively.

## Data Availability

The data presented in this study are available in the Appendix A.

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
