# Peer review of "Transcription Factor Action Orchestrates the Complex Expression Pattern of CRABS CLAW in Arabidopsis"

_genes, 2021, doi:10.3390/genes12111663_

Round 1

Reviewer 1 Report

Authors performed interesting and valuable research concerning isolation and characterization of trans-factors that regulate the CRC promoter activity. Research is generally well planned and performed. Some points may be addressed to improve the manuscript:

Line 134-130: describe if the SALK lines of trans-factors were used to have a mix of trans-factors from different families or were selected under different criteria?

Line 135-136; describe the details of PCR cycling reaction used to isolate the CRC 3.8 kb promoter. Also amount of genomic DNA used as a template, name of polymerase  or polymerase mix used in PCR reaction.

Line 143- provide the lower inhibitory concentration of Aureobasidin, was it the same for all baits?

Line 146- libraries nr 2 and 3 are provided in table S4 not S3- correct it. Probably table S3 contains list of identified trans-factors putatively interacting the CRC promoter.

Library used by Mitsuda et al. is large, contains about 1500 trans-factors. The other two libraries used by Authors were much smaller; why Authors used these libraries ?

Provide if the positive control was used in Y1H test (for example based on p53)- which concentration of aureobasidin was required for positive control?

Line 189- why the r within 1-0.8 and -1 to -0.8 were used?

Lines 244-249- if Authors used 13 bait strains and discarded 4, the it should stay 9 not 10. In line 247 in the bracket Authors show 3 but not four strains that should be discarded-therefore in fact should stay 10. Correct it.

Line 250 – the same comments as in line 146

Line 254- this file (Table S4) contains 2 libraries but not the list of 140 trans-factors putatively interacting with CRC promoter. Correct it.

Author Response

Answer to Reviewer 1   

Line 134-130: describe if the SALK lines of trans-factors were used to have a mix of trans-factors from different families or were selected under different criteria?

SALK lines were chosen to cover a wide range or TF families and due to known involvement in flower development

Line 135-136; describe the details of PCR cycling reaction used to isolate the CRC 3.8 kb promoter. Also amount of genomic DNA used as a template, name of polymerase or polymerase mix used in PCR reaction.    

The Phusion DNA polymerase (Thermo Scientifc) was used to amplify proCRC, using the high fidelity mastermix (Thermo Scientifc) and 5 ng of template DNA.  Initial denaturing at 98 C for 1 min was followed by 40 cycles of 10s 98 C, 20s 58C, 2min 72 C, and a final polishing stept for 5 min at 72C. This is now mentioned in the main manuscript text.

Line 143- provide the lower inhibitory concentration of Aureobasidin, was it the same for all baits?

200 ng/ml was used for all baits, except for the full length bait with 600 ng/ml. We have added this to the main manuscript text.

Line 146- libraries nr 2 and 3 are provided in table S4 not S3- correct it. Probably table S3 contains list of identified trans-factors putatively interacting the CRC promoter.

We have corrected this, thanks for pointing this out.       

Library used by Mitsuda et al. is large, contains about 1500 trans-factors. The other two libraries used by Authors were much smaller; why Authors used these libraries ?

The additional libraries contained TFs which are not included in the Mitsuda library, especially important to us was the inclusion of transcription factors known to be involved in carpel development. Thus we added them as additional libraries. Libraries 2 and 3 were necessesary, as the used prey vectors had either a LEU or a TRP marker

Provide if the positive control was used in Y1H test (for example based on p53)- which concentration of aureobasidin was required for positive control?              

We used the initially the p53-AbAi + pGADT7-p53 positive control (100 ng/ml AbA) and switched later to another positive control (pKCS15-AbAi + pGADT7 CRC, 150 ng/ml AbA). This is now also mentioned in the main manuscript text.

Line 189- why the r within 1-0.8 and -1 to -0.8 were used?            

Correlation values in the range 0.7 - 1 (-0.7 - -1) indicate a strong linear relationship (as summarized in Ratner 2009 (https://doi.org/10.1057/jt.2009.5). We have chosen thus 0.8, respectively -0.8 as cutoff to filter for the highest correlations.

Lines 244-249- if Authors used 13 bait strains and discarded 4, the it should stay 9 not 10.In line 247 in the bracket Authors show 3 but not four strains that should be discarded therefore in fact should stay 10. Correct it.

We have corrected this, thanks for pointing this out.       

Line 250 – the same comments as in line 146      

We have corrected this, thanks for pointing this out.       

Line 254- this file (Table S4) contains 2 libraries but not the list of 140 trans-factors          

putatively interacting with CRC promoter. Correct it.       

We have corrected this, thanks for pointing this out.       

Reviewer 2 Report

In this manuscript on the transcriptional regulation of Crabs Claw (CRC), the authors attempt to piece together what is the dominant mechanisms by which CRC is regulated: epigenetic, miRNA, and by transcription factor binding. In its current form this manuscript is full of errors: no legends for the supplemental, mislabeled figures in the main body, figure legends that don't correspond to the figure, incomplete sentences, etc. The manuscript felt rushed and incomplete, with an overly extensive discussion that I felt was an overinterpretation of the results.However, I have included some specific comments below.

The authors perform several in silico analyses to determine that CRC is probably not regulated by epigenetic/miRNA means, although their epigenetic data is from leaf tissue, and their tissue/organ of interest is the gynoecium. They mention that there are not miRNA binding sites in CRC, but they don't specify what parameters they used with psRNAtarget to demonstrate this negative result.

The above two results lead them to determine which transcription factors are regulating CRC. I thought this was an interesting direction, as there are many publications (which they cite) that have covered this topic (AG is a well known TF that regulates CRC for instance). In addition, the Y1H assay is fraught with caveats, particularly given that mutants of several of the putative regulators did not reveal changes in CRC expression/localization. 

What is the crc-8 background? Figure 1 and Supplemental Figure 4 do not describe this, and if it is newly characterized shouldn't there be a reference? Or actual characterization? Speaking of Figure 1, what is shown in this figure? The colored boxes and lines mean nothing without adequate description in the figure legend.

I can't read all of the TFs in Figure 2 (the first figure 2) but I am surprised that Agamous (AG) is not mentioned as a regulator of CRC. I would have thought that this would be used as a positive control if nothing else (or the MADS box proteins SEP3/4.

How was the "digital gene expression" performed? The authors used public RNA-seq data, but they don't mention how it was normalized, nor how they corrected for batch effects (RNA-seq performed by completely different groups). The methods here are woefully incomplete. In addition, why would the authors think that transcripts that are similarly expressed as CRC would be regulating CRC? The analysis of these data does not make sense to me at all.

The legend for Figure 4 is incorrect.

The discussion is more about other people's results than it is discussing the findings of this paper. Where did the authors demonstrate that CRC is "not only directing auxin synthesis and readout by repressing TRN2... but is also regulated by the auxin response factor ETT" (lines 476-479)? As far as I can tell, TRN2 and YUC4 are not visualized in their heatmap, and are only mentioned in the discussion - not anywhere in the results that I could find.

Round 2

Reviewer 1 Report

Authors corrected the text according to suggestions. I see only small, mainly typographical errors:

Lines 140-141 Authors write C – should be °C.

Line 158- correct the font size and font type as in the other part of text.

Line 184-185; there is no Suppl Fig 6- there are only five Suppl Figures, correct this fragment.

Authors may add the citation of Ratner et al. 2009 to the references list.

Author Response

Authors corrected the text according to suggestions. I see only small, mainly typographical errors:

Lines 140-141 Authors write C – should be °C.

Line 158- correct the font size and font type as in the other part of text.

Line 184-185; there is no Suppl Fig 6- there are only five Suppl Figures, correct this fragment.

Authors may add the citation of Ratner et al. 2009 to the references list.

We have corrected all errors and added the citation to the reference list.

Reviewer 2 Report

I apologize for my terseness in my previous review. I was given even less time to review than normal, and because of confusion with figures and methods I was unable to articulate my concerns. I have attempted here to reword my original concerns.

This version is much improved - having figure legends helps. I will try and expand on a few of my previous concerns, as well as ask some minor points that I can now address.

In terms of the Y1H assay - it is a great assay to use as a screen, but as the authors themselves pointed out, it cannot elaborate on heterodimeric TFs and it is investigating TF-DNA interactions outside of the local chromatin context that would be present in the particular cells of interest. Thus, it is really just a high throughput EMSA. This isn't horrible, but it doesn't indicate that those interactions will take place in the gynoecium. The authors are aware of this, I am just elaborating on my point in my first review.

This gets to my poorly made point about whether these interactions are direct in vivo, in this particular cell type. The novelty of this manuscript is that the authors are looking at CRC regulation in a tissue in which it hasn't already been examined (CRC regulation has been examined elsewhere in the developing flower/floral organs). Thus, whether or not CRC is regulated by one of the many TFs they identify in their Y1H is still unclear - until they have ChIP-seq or some other assay of in vivo interaction (DAP-seq for instance). I am not suggesting more experiments, just indicating that I don't necessarily agree that because Y1H recovered an interaction, and ChIP-seq data are available for a different tissue (from what I could infer), that the same TFs are binding the CRC promoter in this tissue - the authors are arguing that complex regulation requires complex TF interactions - I agree - but we don't know which ones are actually active here.

In terms of co-expression, two very different types of RNA-seq datasets are used here - one from LCM and one from more "bulk" tissues. I think that if the authors were able to normalize their data and correct for the different "batches", they might actually see improved z- scores. As it is, the expression differences they observe could be due to multiple factors. This again, isn't a request for more data, just referring to their discussion point on GRNs (lines 632-644). They are combining "more cell-type specific" with "more broad tissues" expression data. Their summary statement (lines 648-651) could cite one of a number of articles that describe ways to overcome these types of issues (Serin et al., 2016: doi: 10.3389/fpls.2016.00444; Cortijo et al., 2020: doi.org/10.3389/fpls.2020.599464; doi.org/10.1093/abbs/gmz080). 

In terms of the author's response to my question about how genes which are similarly expressed could be regulating one another - I think my concern for this goes back to the incompleteness of the RNA-seq analysis and the lack of knowledge regarding direct interactions in their specific tissue (as mentioned above).

Regarding the discussion - it felt to me that the discussion was less about the findings in this paper and more a review of the field. For instance, "real" (lines 440-442) regulators of CRC are known - extensively. My understanding is that the authors are looking for regulators of CRC in the gynoecium. Thus, why have a paragraph on ATHB5 and ATHB16 repression of CRC in leaves? I think it would be easier to process if the discussion was focused on what was unique about the regulation in the gynoecium versus other parts of the flower (as they list out as rationale for their work in lines 111-116).

Regarding the corrections:

The figure legends were quite helpful. I am still not clear on what the colored boxes in Figure 1 are meant to be. 

What was the rationale for using Ler-0 as the source of genomic DNA for Y1H? Are there regulatory differences between Ler-0 and Col-0 (the source of crc-8) that could be useful for examining TF binding sites? Could this have anything to do with observed differences in results with prior literature? 

I think this is probably how the figures were inserted into the mock publication for review (by the journal) but they are still quite difficult to read - particularly Figure 2.

The authors emphasize that the rows have not been correlated in Figure 3, but a clustering algorithm has clearly been run because of the dendrogram to the left of the heat map. I think this might create confusion to the reader - what exactly is not correlated if they are clustering similar patterns?
